# A Study on MDA5 Signaling in Splenic B Cells from an Imiquimod-Induced Lupus Mouse Model with Proteomics

**DOI:** 10.3390/cells11213350

**Published:** 2022-10-24

**Authors:** Yu-Jih Su, Fu-An Li, Jim Jinn-Chyuan Sheu, Sung-Chou Li, Shao-Wen Weng, Feng-Chih Shen, Yen-Hsiang Chang, Huan-Yuan Chen, Chia-Wei Liou, Tsu-Kung Lin, Jiin-Haur Chuang, Pei-Wen Wang

**Affiliations:** 1Department of Internal Medicine, Kaohsiung Chang Gung Memorial Hospital, Chang Gung University College of Medicine, Kaohsiung 83301, Taiwan; 2Center for Mitochondrial Research and Medicine, Kaohsiung Chang Gung Memorial Hospital, Chang Gung University College of Medicine, Kaohsiung 83301, Taiwan; 3Departments of Nuclear Medicine, Kaohsiung Chang Gung Memorial Hospital, Chang Gung University College of Medicine, Kaohsiung 83301, Taiwan; 4Institute of Biomedical Sciences, Academia Sinica, Taipei 11529, Taiwan; 5Institute of Biomedical Sciences, National Sun Yat-Sen University, Kaohsiung 804201, Taiwan; 6Medical Research and Core Laboratory for Phenomics and Diagnostics, Kaohsiung Chang Gung Memorial Hospital, Chang Gung University College of Medicine, Kaohsiung 83301, Taiwan; 7Department of Neurology, Kaohsiung Chang Gung Memorial Hospital, Chang Gung University College of Medicine, Kaohsiung 83301, Taiwan; 8Pediatric Surgery, Kaohsiung Chang Gung Memorial Hospital, Chang Gung University College of Medicine, Kaohsiung 83301, Taiwan

**Keywords:** MDA5, lupus, B cell, splenocyte, proteomics

## Abstract

Introduction: Several environmental stimuli may influence lupus, particularly viral infections. In this study, we used an imiquimod-induced lupus mouse model focused on the TLR7 pathway and proteomics analysis to determine the specific pathway related to viral infection and the related protein expressions in splenic B cells to obtain insight into B-cell responses to viral infection in the lupus model. Materials and Methods: We treated FVB/N wild-type mice with imiquimod for 8 weeks to induce lupus symptoms and signs, retrieved splenocytes, selected B cells, and conducted the proteomic analysis. The B cells were co-cultured with CD40L+ feeder cells for another week before performing Western blot analysis. Panther pathway analysis was used to disclose the pathways activated and the protein–protein interactome was analyzed by the STRING database in this lupus murine model. Results: The lupus model was well established and well demonstrated with serology evidence and pathology proof of lupus-mimicking organ damage. Proteomics data of splenic B cells revealed that the most important activated pathways (fold enrichment > 100) demonstrated positive regulation of the MDA5 signaling pathway, negative regulation of IP-10 production, negative regulation of chemokine (C-X-C motif) ligand 2 production, and positive regulation of the RIG-I signaling pathway. A unique protein–protein interactome containing 10 genes was discovered, within which ISG15, IFIH1, IFIT1, DDX60, and DHX58 were demonstrated to be downstream effectors of MDA5 signaling. Finally, we found B-cell intracellular cytosolic proteins via Western blot experiment and continued to observe MDA5-related pathway activation. Conclusion: In this experiment, we confirmed that the B cells in the lupus murine model focusing on the TLR7 pathway were activated through the MDA5 signaling pathway, an important RNA sensor implicated in the detection of viral infections and autoimmunity. The MDA5 agonist/antagonist RNAs and the detailed molecular interactions within B cells are worthy of further investigation for lupus therapy.

## 1. Introduction

For systemic lupus erythematosus (SLE), approximately 40 to 50 predisposing genes were identified in recent genome-wide association studies of various racial groups. Such genes confer hazard ratios for SLE of 1.5–3 and account for approximately ten to twenty percent of disease susceptibility, thus indicating that environmental exposure and epigenetics also play major roles [1]. Antigen-presenting human leukocyte antigen molecules play a crucial role in antigen presentation and genetic diversity was observed. Other genes in innate immunity pathway gene polymorphisms, genes in lymphocyte signaling pathways, pathways that affect the clearance of apoptotic cells [2] or immune complexes, genes that influence neutrophil adherence, and genes that influence DNA repair have all been associated with SLE [3]. Several environmental stimuli may influence SLE, particularly viral infection [4,5]. The Epstein–Barr virus (EBV) may be one such infectious agent that can trigger SLE in susceptible individuals [6,7], and children and adults with SLE are more likely to be infected by EBV than age-, sex-, and ethnicity-matched controls. EBV enhances the TLR7 pathway in humans [8] and drives the T helper 17 cell-related pathway in the lupus mouse model [9]. Furthermore, a recent CRISPR-Cas9 experiment clearly delineated the role of cxorf21 in female-bias immune response in SLE [10]. Therefore, the interplay between genetic susceptibility, environment, gender, and abnormal immune responses can result in autoimmunity.

SLE is often treated with steroids or other cytotoxic drugs, often in combination with several other immunosuppressants due to major organ involvement or treatment resistance. Only one biological agent has been approved for treating SLE in the past 20 years [11], and that agent targets the B-cell pathway, indicating that the B-cell-specific pathway is a gateway for treating SLE [12]. Therefore, we focused on B-cell research in this study. We observed that viral infections have tremendous effects on SLE patients. In our previous clinical study and observations, our team discovered that cytomegalovirus activity was correlated with SLE disease activity [5]. The interaction between SLE disease activity status and the virus was a specific research interest of our team [13], while several other research articles have focused on the viral sensing molecules inside the cells and in the disease, particularly in SLE [14]. More than ten years ago, focusing on the modification of viral sensing molecules with different novel DNA or RNA became a new strategy for targeting SLE [15]. In fact, SLE is a syndrome of interferonopathy [16], so different immune cells may contribute to the significant pathogenesis of SLE, including dendritic cells [17], T cells [18], and B cells [19]. Recently, gene-centered DNA sequencing in patients with SLE demonstrated that the B-cell receptor signaling pathways are significantly up-regulated among the SLICC damage index in SLE patients [20]. 

The type 1 interferon (IFN) pathway, a classic anti-viral signaling pathway, was recognized in the pathogenesis of SLE [21]. The molecules operating upstream of type 1 IFN production, including the endosomal TLRs [22] and the RNA helicase pathway [23], play an important role in disease susceptibility. The three major intracellular proteins of the helicase pathway are Melanoma Differentiation Associated gene 5 (MDA5), Retinoic acid Inducible Gene I (RIG-I), and Laboratory of Genetics and Physiology 2 (LGP2), which orchestrate each other within the overlapping downstream pathway [24,25]. Activation of RIG-I and MDA5 by nuclear acid leads to the binding of MAVS on the mitochondrial membrane and then initiates the cascade of NF-κB, IRF3, type 1 IFN, and pro-inflammatory cytokines [26]. Previously, we found that MDA5 is higher in lupus patients than in disease controls [2], and the downstream molecule on the inner mitochondrial membrane, MAVS, is significantly negatively associated with disease activity [2]. 

The aim of this study is to utilize a model mimicking viral infection that targets TLR7, to determine the B cells’ role in lupus [4,21] and check the role of the mitochondrion in B-cell activity [5]. We utilized the proteomics analysis to determine the specific pathway activation and its related protein expressions to guide the future development of SLE therapeutics. 

## 2. Material and Methods

### 2.1. Mouse Study

To establish the phenotype of SLE expression in FVB/N wild-type mice, we followed the protocol described in a previously published article [27]. The lupus mouse model was developed by stimulating the foreign nucleic acids receptor TLR7 using the chemical imiquimod. Mice were bred within the Association for Assessment and Accreditation of Laboratory Animal Care certified (AAALAC International certified) center. Each experiment contained at least three separate animals. All the experiments showed similar and comparable results. The histology experiments were also performed with three different staining methods and the results were all comparable. Briefly, the FVB/N wild-type 7-week-old male mice were divided into two subgroups, control versus IMQ-treated, and each subgroup included at least three mice. The control subgroup mice were not treated with anything but the IMQ-treated subgroup mice had topical IMQ (125 mg, 5% IMQ, Aldara^TM^ cream, MEDA) applied every other day during the weekday for seven successive weeks to induce SLE before following with the experiments. We then examined the phenotypes of SLE in different organs in all experimental mice. Serum was collected when the mice were sacrificed. In total, nine mice from the FVB/N subgroup and 13 mice from the FVB/N + IMQ subgroup performed with creatinine (Cre), blood urea nitrogen (BUN) and glutamate pyruvate transaminase (GPT) detection. When the mice were sacrificed, we extracted their urine from the bladder and detected protein and glucose concentrations with urine test strips. We also collected the ascites and tested them with test strips. All the animal experiment protocols were reviewed and approved by Chang Gung Memorial Hospital Experimental Animal Care and Usage Committee, No. 2019093002.

### 2.2. Pathological Analysis of Mouse Kidney, Liver, and Spleen

Tissues from the mouse spleen, liver, and kidney were obtained, fixed in 4% paraformaldehyde, and embedded in paraffin. Tissues were sectioned at 4 µm thickness, dewaxed using xylene, dehydrated through a gradient of alcohol, and stained with hematoxylin and eosin (H&E). We analyzed the proteinuria as follows: Urine was collected weekly and protein was screened via dipstick followed by albumin measurements via Albuwell kits (Exocell, Philadelphia, PA, USA) and total Urine creatinine via commercial kit (BioAssay Systems, Hayward, CA, USA). We represented urinary protein excretion using the albumin/creatinine ratio.

### 2.3. Flowcytometry Gating Strategy and Serology or Urine Analysis

First, we gated lymphocytes and then distinguished dead and live cells using 7-AAD staining, and for the gating of live cells, we drew a cross-quadrant diagram. We used CD3 as a T-cell marker and CD19 as a B-cell marker. The percentage is the ratio of the number of cells in the gating area to the total number of cells charged.

Serology analysis with anti-dsDNA (anti-dsDNA) total Ig (IgG, IgA, IgM) in serum was detected by using ELISA kits from Alpha Diagnostic, 5110. The experimental method was performed according to the manufacturer’s instructions.

When the mice were sacrificed, urine was drawn from the bladder, and urine protein and urine glucose were immediately measured with urine test strips (Urisrix, SIEMENS), and their scores were defined as follows: negative: 0, +: 1, ++: 2, +++: 3, and ++++: 4 (Figure 1C). At the time of sacrifice, mouse heart blood was collected, and its serum was preserved, and creatinine (mg/dL) and blood urea nitrogen (BUN, mg/dL) in serum were measured with an automatic dry biochemical analyzer (Spotchem EZ SP-4430, Arkray, Shanghai, China) and glutamate pyruvate transaminase (GPT, IU/L); the results are shown in Figure 1B.

### 2.4. B Cells Isolated from Mouse Splenocytes and Cultured with Feeder Cell

Splenocytes were isolated and co-cultured with feeder cells. Mice spleens were placed in MACS buffer (Miltenyi Biotec, 130-091-221) and the spleen was ground with a 5 mL syringe on a 100 µm mesh (CORNING, 352360) until no solid tissue could be seen. The supernatant was removed by centrifugation and 3 mL RBC lysis Buffer (RBC bioscience, S31100999) was added to remove red blood cells; then the cell block was centrifuged and washed twice with PBS to redissolve in MACS buffer in order to obtain the splenocyte suspension. 

B cells were purified from splenocytes and cultured with feeder cells. Unwanted cells in the splenocytes were targeted for removal with biotinylated antibodies directed against non-pan-B cells (CD4, CD8, CD43, CD49b, TER119) and streptavidin Particles Plus-DM (BD IMag™, 557812). We separated labeled cells using a Cell Separation Magnet (horizontal position) without the use of columns and the desired cells were poured into a new tube. After counting the cells, 5 × 106 cells were taken to confirm the purification effect by flow cytometry; Co-Culture 3 × 10^4^ B cells into a 6-well plate with 6 × 10^4^ cell/well feeder cell; after 7 days of co-culture, CD19-APC (BD pharmingen, 555415) and APC magnetic particles (BD IMag™, 557932) were used to separate the B cells.

The feeder cell was a generous gift from Professor Garnett Kelsoe, Professor David Baltimore, and Dr. Kuei-Ying Su [28].

### 2.5. Proteomics Analysis

Following the aforementioned negative selection methods of B cells, we then positively selected B cells with CD19 surface markers for proteomics analysis. Splenocyte B cell lysates (50 µg) collected from imiquimod-induced lupus mice at day 28 post-induction were dissolved in urea, reduced with 20 mM dithiothreitol at 56 °C for one hour and alkylated with 55 mM iodoacetamide at 25 °C for three quarters. Then, Trypsin and Lys-C Mix (Promega, Madison, WI, USA) were added to the above samples at a ratio of 50:1 (protein/protease, *w*/*w*) and incubated at 37 °C for another four hours. Protein samples were then diluted to a final concentration of 1 M urea with triethylammonium bicarbonate at 50 mM and incubated at 37 °C for another 17 h. The digested samples were acidified with formic acid to pH = 2, desalted with C18 Oasis^®^PRiME HLB cartridges (Waters, Milford, MA, USA), and subjected to tandem mass tag (TMT) labeling using TMT6-plexTM reagents (Thermo Scientific, Waltham, MA, USA.). The reagents were dissolved in 41 µL of anhydrous acetonitrile and then added to 100 µg of peptides dissolved in 100 µL of triethylammonium bicarbonate at 50 mM. After another hour of incubation, the reaction was quenched by 8 µL of 5% hydroxylamine and incubated for a quarter at 37 °C. Labeled peptides from the samples were combined and desalted with C18 Oasis^®^ PriME HLB cartridges. The combined TMT-labeled peptides were solubilized in buffer A (20 mM ammonium formate, pH 10) and separated on an Xbridge BEH130 C18 column (3.5 µm, 2.1 × 150 mm, Waters, USA) using an 1100 series HPLC equipped with a UV detector (Agilent, USA). The separation was performed with a fixed 90 gradient, at 5% buffer B (20 mM ammonium formate with 80% acetonitrile, pH 10) for 5 min, linear increase from 5 to 43% B over 45 min, followed by a linear increase to 100% B over 20 min, and then isocratic at 100% for 5 min before re-equilibration at 5% B for 15 min. The TMT-labeled peptides were dissolved in 0.1% aqueous formic acid solution and relatively quantified by LC–MS/MS on an Easy-nLC 1200 system (Thermo Scientific, USA) coupled with an Orbitrap Elite™ Mass Spectrometer (Thermo Scientific, USA). Full MS (*m*/*z* 350–1600) resolutions were set to 120,000 and MS2 scans were acquired using the settings according to the manual’s settings. The raw files were processed with Proteome Discoverer v.2.4 (Thermo Scientific, USA) for protein identification and TMT-based relative protein quantification. A database search was performed through the Sequest HT against the mouse proteome FASTA files (2020-10, 63,702 entries) downloaded from the UniProt database. We verified the peptide spectrum matches by q-values (1% false discovery rate) from the Percolator algorithm in the Proteome Discoverer based on a decoy database search. Protein quantification was accomplished by assessing the relative signal-to-noise values of the reporter ions extracted from MS2 spectra. Two separate B-cell samples from two different animals were obtained and sent for proteomics analysis. Proteomic experiments were performed a total of four times, including two control experiments and two imiquimod-stimulated B cells.

### 2.6. Western Blot Confirmation of Intracellular Proteins after Feeder Cells Co-Culture and Then after Nuclear–Plasma Separation of B Cells

Cells were divided into two equal aliquots, one of which was resuspended in 500 µL of 50 µM NaOH and boiled for 30 min to solubilize DNA. A total of 50 µL of 1 M Tris-HCl pH 8 was added to neutralize the pH and these extracts served as normalization controls for total mtDNA. The second equal aliquot was resuspended in roughly 500 µL buffer containing 150 mM NaCl, 50 mM HEPES pH 7.4, and 25 µg/mL digitonin (Sigma, St. Louis, MO, USA). The homogenates were incubated end over end for 10 min to allow for selective plasma membrane permeabilization and then centrifuged three times at 980× *g* for 3 min in order to pellet intact cells. The cytosolic supernatants were transferred to fresh tubes and spun at 17,000× *g* for 10 min to pellet any remaining cellular debris, yielding cytosolic preparations free of nuclear, mitochondrial, and endoplasmic reticulum contamination. The purity of the cytosolic fractions was tested using Western blot; Lamin A/C was probed as nuclear loading control; Calnexin was probed for endoplasmic reticulum; COX IV was probed for mitochondria; and β-actin was probed as a cytoplasmic control. The proteins extracted from B cells were dissolved using the protein extraction reagent (#78510, Thermo Fisher Scientific, Carlsbad, CA, USA). We used the primary antibody for target protein and HRP-conjugated secondary antibody. We quantified the amount of detected protein using ImageJ software and expressed it as the ratio to β-actin protein. Primary antibodies against mouse cGAS, RIG-1, MDA5, MyD88, TRAF6, and IRF3 were utilized to demonstrate cytosolic protein concentrations. 

We used the Nuclear Extraction Kit (Chemicon Internation, 2900) for nucleocytoplasmic separation, quantified protein with NanoDrop ND-1000, and denatured samples for 5 min at 100 °C in the presence of 4× Sample loading Buffer and reducing agent (Invitrogen, Carlsbad, CA, USA). Fifty micrograms of samples were separated by SDS/PAGE on 10% gels. Each gel was initially run for 30 min at 80 V and then at 110 V. Transfer onto nitrocellulose membranes (BioRad, Hercules, CA, USA) was performed using a Trans-Blot Turbo Transfer system. Membranes were blocked for 1 h with 5% BSA (Sigma Aldrich) at room temperature in PBS supplemented with 0.05% Tween-20 (PBST). Membranes were probed overnight at 4 °C with the following primary antibodies: cGAS (CST, 31659s), RIG-1 (CST, 3743s), MDA5 (CST, 5321s), MyD88 (Santa Cruz Biotechnology, Santa Cruz, CA, USA, sc-11356), TRAF6 (Santa Cruz Biotechnology, sc-8409), IRF3 (Abcam, Boston, MA, USA, ab68481), and secondary antibody Goat anti-rabbit IgG (H + L) HRP (Thermo Fisher Scientific, 32460). All membranes were washed with PBST and exposed using the SuperSignal West Pico PLUS chemiluminescent substrate (Thermo Fisher Scientific). We adopted Image J software to quantify the Western blot results.

### 2.7. Statistical Analysis

Results are expressed as mean ± standard deviation. Between-group and within-group comparisons were performed using the unpaired and paired *t*-tests, respectively. All statistical operations were performed using the Statistical Package for Social Science program (SPSS version 22 for Windows). A *p*-value < 0.05 was considered statistically significant. Furthermore, using online Panther analysis with GO pathway analysis, we uploaded those proteins that differed statistically from the baseline (*p* < 0.05) and chose the GO biological process complete with Fisher’s exact test and the Calculate False Discovery Rate to compare them with the Reference List, which includes all genes from a whole genome of Mus musculus genes provided by Panther [29]. The protein–protein interactome was analyzed using the STRING database data, which is available online since 23 June 2022 at https://string-db.org/.

## 3. Results 

### 3.1. Successful Induction of Lupus Presentation in Mice with Imiquimod Demonstrated by Histology and Serology

The FVB/N mice stimulated with IMQ were shown to have oval-shaped abdominal swelling, with hepato-splenomegaly and ascites found (Figure 1A). The animal study results of the biochemistry tests and the pathological changes after stimulation with IMQ for 7 weeks are shown in Figure 1. It can be clearly demonstrated from the appearance that the abdomen of the experimental group (*n* = 10) is enlarged compared with the control group (*n* = 9) and the liver and spleen are also noticeably enlarged (arrows) (Figure 1A). The addition of hepato-splenomegaly suggests liver diseases in IMQ-treated mice, even though the GPT levels were comparable between FVB/N wild-type and IMQ-treated mice (Figure 1B). After the internal organs were removed, the pancreas of the experimental group was also clearly enlarged compared to the control group. The creatinine (Cre), blood urea nitrogen (BUN) and glutamate pyruvate transaminase (GPT) results showed no significant difference between the two groups (Figure 1B). The concentration of urinary protein of the experimental group (*n* = 14) was higher than the control group (*n* = 7) (*p* = 0.0053) but no significant difference was observed in urine glucose (*p* = 0.1626). Some mice in the experimental group were found to have ascites when they were sacrificed (6 of 15). Higher concentrations of glucose were detected in ascites fluid. Since no mice in the control group developed ascites, a comparison between the two groups could not be made (Figure 1C). The ascites were found to have a high glucose concentration (over 2000 mg/dL) and medium protein levels (around 1 mg/dL), thus indicating transudative ascites (Figure 1C). The H&E staining over the liver and kidney tissues showed lymphocyte infiltration over the portal vein area in the liver, as well as interstitial in the glomerular area and perivascular in the kidney section (shown as boxed, Figure 1D).

### 3.2. B Cells from Spleen via Negative Selection Method Demonstrated Activation and Proliferation with Pathology and Intracellular Protein Analysis

The germinal centers and the whole spleen were hyperactively enlarged in those mice stimulated with IMQ (right side photos in Figure 2A) compared to those mice without IMQ (left side photos in Figure 2A). We purified B cells for further culture system but, prior to culture, we checked the surface expression markers of CD3 and CD19 by flowcytometry, as demonstrated in Figure 2B. Figure 2B includes four panels of the flow cytometry analysis from the splenocytes of a mouse. The left two panels are splenocytes before purification and the data are demonstrated with one of three individual experiments. These three different experiments have comparable results. The said result shows that the CD19(+)B cells are 46.9% of total splenocytes and T cells are 15.3% of splenocytes in the FVB/N mouse in the left upper panel; the CD19(+)B cells are 16.5% of total splenocytes and T cells are 13.7% of splenocytes in the FVB/N + IMQ mouse in the left lower panel. The right two panels are B cells after negative B-cell selection (purification process) from the splenocytes. The result shows that the CD19(+)B cells are 81.4% of total splenocytes and T cells are 0.4% of splenocytes in the FVB/N mouse in the right upper panel; the CD19(+)B cells are 30.5% of total splenocytes and T cells are 1.6% of splenocytes in the FVB/N + IMQ mouse in the right lower panel (Figure 2B).

Figure 2C is the histogram of Figure 2B. The X-axis is the grouping, and the Y-axis is the proportion of B cells in the total number of cells collected. The FVB/N group had an average of 45.9% before purification and an average of 77.48% after purification; the FVB/N + IMQ group had an average of 28.58% before purification and an average of 45.36% after purification (*n* = 3 per group, * indicates *p*-value < 0.05).

The culture result is demonstrated in Figure 2D. The photo from day 1 is shown in the photo on the left and the photo of day 7 is shown on the right of Figure 2D. The B-cell numbers were counted in the same high-power field to compare the numbers between day 1 and day 7 of B-cell cultures. The data are demonstrated in Figure 2E in the histogram.

B cells secreting the autoantibody between those B cells from IMQ-stimulated mice and those B cells from non-stimulated mice are demonstrated with anti-dsDNA autoantibody secretion as determined by the ELISA kit in Figure 2F.

The Western blot results of intracellular proteins inside the B cells are shown. Those B cells, either from IMQ-stimulated or non-stimulated mice, were co-cultured with CD40L feeder cells for 7 days and were then retrieved for analysis. Cytosolic protein was separated using the centrifuge method and blotted with antibodies against mouse cGAS, RIG-1, MDA5, MyD88, TRAF6, and IRF3, as shown in the upper part of Figure 3. The histograms are also demonstrated in the lower part of Figure 3.

### 3.3. Proteomics Results Comparing B Cells from the Spleen of Imiquimod-Stimulated FBV/N Mice with the Reference from the Panther Classification System [29] and the STRING Database

Proteins differentially expressed by B cells were compared to Mus musculus in the Panther classification system database [25]. We selected those proteins available from the proteomics analysis with a *p*-value of less than 0.05 and 1.5 times changed from the baseline from mice before and after stimulation with IMQ. 

The mass spectrometry proteomics data were deposited to the ProteomeXchange Consortium via the PRIDE partner repository (Website: http://www.ebi.ac.uk/pride) (accessed on 23 June 2022) with the dataset identifier PXD034855.

Among those up-regulated genes, 34 annotated genes showed statistically significant differences in fold-change (*p*-values < 0.05 by *t*-test) (Table 1). Among the 34 genes, 27 were expressed inside the cytosol and were involved in protein–protein interaction analysis (Figure 4a). A unique interactome containing 10 genes was discovered by the STRING protein–protein interactome database and said interactome was found to be involved in RIG-I-like receptor signaling, pyrimidine metabolism, and interferon-inducible GTPase pathways (Figure 4b). Within this interactome, ISG15, IFIH1 (also known as MDA5), IFIT1, DDX60, and DHX58 were previously demonstrated to be downstream effectors of MDA5 signaling. We adopted a volcano plot to show the proteins differentially expressed in B cells from either IMQ-stimulated or non-stimulated mice (Figure 4c).

All the proteins are listed in Table 1, which is comparable to the Mus musculus in the Panther classification pathway analysis [29], and the results demonstrated the positive regulation of the MDA5 signaling pathway with the highest fold enrichment over 100. The pathways activated with these proteins are shown in Figure 5.

### 3.4. Western Blot Confirmation of Intracellular Proteins of B Cells after CD40L+ Feeder Cells Co-Culture for 7 Days

To confirm our findings from the proteomics, we checked the activated pathways inside B cells before and after IMQ stimulation following nuclear–plasma separation to determine the role of MDA5-related pathways within the B cells (Figure 5). The GO pathway analysis with those proteins input (Table 1) clearly demonstrates the intracellular signaling pathways of the listed B cells listed (Figure 5).

## 4. Discussion

In the current study, we demonstrate an informative activation cascade inducing lupus serological and pathological findings inside B cells from splenocytes after TLR7 epicutaneous stimulation. With the help of proteomics analysis of B cells from splenocytes and subsequent confirmation via Western blotting, the intracellular activation cascade of B cells was revealed. After TLR7 stimulation, not only were downstream MyD88 and TRAF6 activated, but MDA5 and cGAS proteins were also greatly expressed (Figure 5). The combination of these intracellular signaling pathways propagates B-cell activation and increases anti-dsDNA in the lupus mouse model (Figure 2). The activation of both MDA5 (through dsRNA) and cGAS (through dsDNA) pathways inside the B cells in the spleen following induction may be reasonably explained by the endogenous damage-associated molecular patterns (DAMPs) released from the damaged mitochondria that activate the innate immune system by interacting with pattern recognition receptors (PRRs). Inside B cells, autoimmunity is likely triggered by the simultaneous release of mitochondrial double-strand DNA and mitochondrial double-strand RNA due to an oxidative burst and membrane disruption of the mitochondria.

Proteomics data of splenic B cells in our study revealed that the most important pathway activated is the positive regulation of the MDA5 signaling pathway. We understand that the MDA5 pathway recognizes dsRNA, usually through exogenous dsRNA viral infection [30]. In our experiment, the only source of dsRNA was what had escaped from the mitochondria. We previously demonstrated the release of dsRNA from mitochondria in neuroblastoma cells in a previous publication [31]. 

Antibodies to RNA- and DNA-containing autoantigens are characteristic of systemic lupus erythematosus (SLE). Chauhan et al. reported that the presence of anti-ENA and anti-dsDNA autoantibodies in SLE patients was associated with elevated TLR7 and TLR9 levels [32]. With regard to TLR7 and TLR9, similar findings were reported by Christensen et al., who further mentioned that TLR9 and TLR7 also had dramatic effects on clinical disease in lupus-prone mice [33]. In the absence of TLR9, autoimmune disease was exacerbated. In contrast, TLR7-deficient mice had ameliorated disease, decreased lymphocyte activation, and decreased serum IgG [33,34]. TLR7 is considered to play a pivotal role in a wide variety of autoimmune responses against DNA- and RNA-containing nuclear antigens [35]. However, Hanten et al. studied purified human CD19+ B cells from peripheral blood stimulated with either TLR7-selective agonist, TLR7/8 agonist, or TLR9 selective agonist and found that, despite their molecular differences, the TLR7 and TLR9 agonists induced similar genes and proteins in purified human B cells [36]. 

In the present study, we demonstrated that TLR7 stimulation is linked to anti-dsDNA production in our mouse model, which summarizes the findings from Hanten et al. [36] and Barrat et al. [37]. Although the findings of Hanten et al. hint that the TLR7 and TLR9 agonists may induce similar genes and proteins in B cells, which have an intrinsic expression of both receptors, the absence of anti-ENA antibodies in our experiment otherwise demonstrated the specificity of TLR7 stimulation of IMQ (or another ssRNA) and anti-dsDNA production. In addition, intracellular pathways and extracellular molecules may also contribute to autoantibody production [25,26]. Furthermore, TLR9-deficient mice, excluding the role of TLR9 in the inhibition of TLR7-mediated responses [37], show high lupus activity compared to WT mice. Since a previous study has shown that TLR7 and TLR9 responsive B cells share phenotypic and genetic characteristics [38], some proteins [39] may differentially modulate the activity of either TLR7 or TLR9 in contributing to the downstream antibody production of anti-dsDNA or anti-ENA. As a result, the TLR9 pathway may not be able to efficiently inhibit the TLR7 activation pathway in our model and TLR9 merits further investigation.

The novelty of this research is that we specifically determined the intracellular signaling pathway in B cells after TLR7 stimulation with IMQ by using proteomics analysis in the SLE animal model in order to determine the pathological role of B cells in SLE patients after viral infection. The stimulation of TLR7 with IMQ eventually activated the MDA5 pathway inside the B cells without real viral infection in this model, indicating that the autoimmunity in B cells could be induced by inorganic chemicals in the environment.

With the results of the proteomics from B cells in the current study, we further determined several pathways activated inside the B cells other than the MDA5-associated pathway and found that such pathways were mostly associated with cell cycle progression. For example, pathways associated with mitosis, protein phosphorylation, DNA replication, formation of 40S subunits, G1-S phase transition, SRP-dependent protein targeting to the membrane, and G2-M phase transition were shown in the B cells (data not shown because these pathways need to be validated). These pathways gave us a new direction to determine the status of B cells in relation to disease status. Activation of an intracellular pathway or cell cycle inside B cells could be a hint to particular cellular function and inflammatory status but requires further investigation. 

In summary, our study divided the maturation of B-cell processes into two steps: first was the induction of B cells in vivo by stimulation of mouse skin with the TLR7 agonist, e.g., IMQ, after which we retrieved the splenocytes and purified B cells from the splenocytes with the negative selection according to the protocol; the second step was to co-culturing the purified B cells with the feeder cells according to the protocol, without further antigen stimulation, using only the cytokines necessary for B-cell maturation. We found that both the proteomics data and the pathway analysis point to the MDA5 pathway (shown in Table 1 and Figure 3 before CD40L co-culture). We understand that the MDA5 pathway is the pathway that recognizes dsRNA but the only dsRNA present in this experiment was the mitochondrial-released dsRNA. Otherwise, only viruses form dsRNA in the process of replication [40,41] and the most known RNA inside a host is single-stranded RNA [34]. We previously demonstrated the release of dsRNA from mitochondria in another publication [31]. The stimulation with IMQ and activation of the TLR7 receptor in mice, leading to the mitochondrial activation and release of dsRNA in B cells, is well demonstrated in our proteomics analysis (Table 1) and pathway analysis (Figure 3). 

In the current study, we provide an explanation of B-cell autoimmunity that is activated through the mitochondria-dependent pathway, possibly via releasing mitochondrial dsDNA and dsRNA to further activate downstream intracellular MDA5 and cGAS, as demonstrated by Western blot and proteomics analysis inside this animal model. According to our GSEA analysis, the B cell is activated, as is the ataxia telangiectasia mutated gene in several pathways to demonstrate overactivity of the B cells with DNA breaks. This pathway may also be present in human lupus disease and the detailed molecular interactions inside B cells are worthy of further investigation for future clinical applications.

## 5. Conclusions

The current study demonstrates the molecular mechanism in the imiquimod-induced lupus mouse model inside the B cells of the spleen. The activation of B cells in this model by imiquimod stimulation turned B cells into MDA5 pathway-activated B cells, suggesting that the MDA5-dependent pathway and mitochondria play important roles in the pathogenesis of lupus. Further investigation of the MDA5 agonists/antagonist RNAs and detailed molecular interactions inside B cells is needed for future lupus therapy.

## Figures and Tables

**Figure 1 cells-11-03350-f001:**
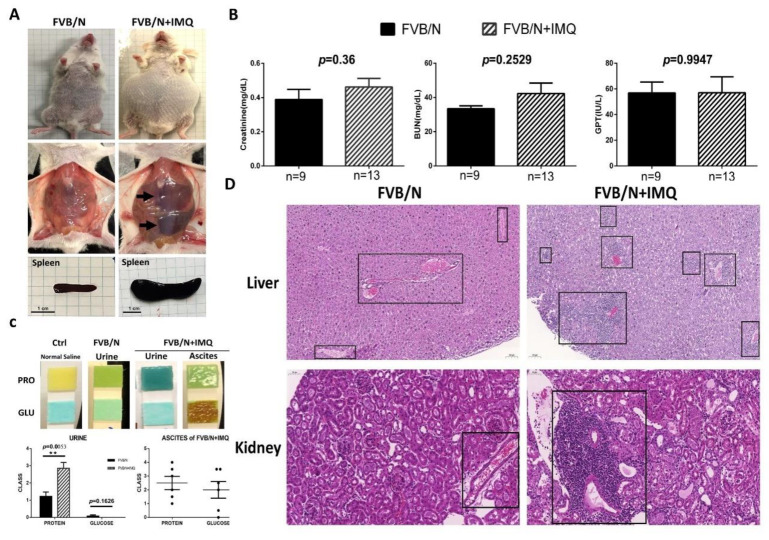
The animal study results of the biochemistry tests and the pathological changes in hematoxylin and eosin staining (H&E) in the two subgroups, control (FVB/N) versus mice treated with IMQ (FVB/N + IMQ). (**A**) The general appearance of the mouse. The abdominal cavity shows hepatosplenomegaly and spleens were extracted to demonstrate the actual size differences between the two subgroups. (**B**) The biochemistry levels of kidney function (Creatinine, BUN, mg/dL) and liver function (glutamate pyruvate transaminase, GPT, IU/L) were demonstrated. No statistical differences are shown and *n* number is indicated in each subgroup. All *p*-values > 0.05. (**C**) The photos demonstrate the protein and glucose levels in each subgroup, and *n* number is indicated in each subgroup as shown in (**B**). The histogram is also demonstrated with a semi-quantitative method. The *p*-value was obtained using unpaired *t*-test by comparing the two subgroups and the result was 0.0053. The ** indicates that the *p*-value was less than 0.01. (**D**) The H&E staining over the liver and kidney tissues, left panels are from one of the three FVB/N mice, and the right panels are from one of the three FVB/N + IMQ mice. The boxed area highlights the lymphocyte infiltration over the portal vein area in the liver, as well as interstitial and the glomerular area in the kidney section.

**Figure 2 cells-11-03350-f002:**
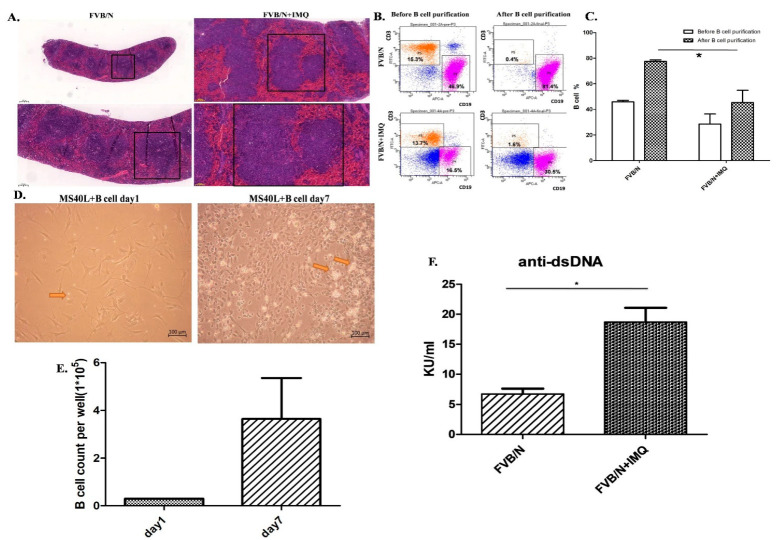
Spleen and B-cell purification from the splenocytes from the mice before and after IMQ stimulation. B-cell function differences are demonstrated with anti-dsDNA autoantibody secretion as determined by ELISA kit. (**A**) The H&E stain of the FVB/N group and the FVB/N + IMQ mouse spleen is demonstrated with one of three experiments on three different mice specimens. The germinal centers of FVB/N + IMQ mouse spleen show enlargement, cell proliferation, and hyperactivation. The left two panels are pictures of 50-times and 100-times magnifications of the original FVB/N spleen specimens. The right two panels are pictures of 50-times and 100-times magnifications of the original FVB/N + IMQ spleen specimens. (**B**) The flow cytometry analysis from mouse splenocytes. The left two panels are splenocytes, and the data are demonstrated with one of three individual experiments. These three different experiments have comparable results. The results show that the CD19(+)B cells are 46.9% of total splenocytes and T cells are 15.3% of splenocytes in FVB/N mouse; the CD19(+)B cells are 16.5% of total splenocytes and T cells are 13.7% of splenocytes in FVB/N + IMQ mouse. The right two panels are B cells after negative B-cell selection from the splenocytes. The results show that the CD19(+)B cells are 81.4% of total splenocytes and T cells are 0.4% of splenocytes in FVB/N mouse; the CD19(+)B cells are 30.5% of total splenocytes and T cells are 1.6% of splenocytes in FVB/N + IMQ mouse. (**C**) Histogram of (**B**) The X-axis is the grouping, and the Y-axis is the proportion of B cells in the total number of cells collected. The FVB/N group had an average of 45.9% before purification and an average of 77.48% after purification; the FVB/N + IMQ group had an average of 28.58% before purification and an average of 45.36% after purification (*n* = 3 per group, * indicates *p*-value < 0.05). (**D**) The picture taken on day 1 (right panel) and day 7 (left panel) from the B cells co-cultured with the CD40L feeder cells. The arrow indicates B cell. (**E**) The number of B cells per well in 6-well plate after 7 days of culture. On the first day, 3 × 10^4^ B cell/well in 6-well plate was fixed and on the 7th day, the average was 3.64 × 10^5^ B cell/well in a 6-well plate (*n* = 3 per group, * indicates *p*-value < 0.05). (**F**) The titers of the anti-dsDNA autoantibody were determined by ELISA for four different individual experiments in each group. Data are expressed as mean ± standard deviation and comparison was performed by Mann–Whitney test. The mark * indicates a *p*-value less than 0.05.

**Figure 3 cells-11-03350-f003:**
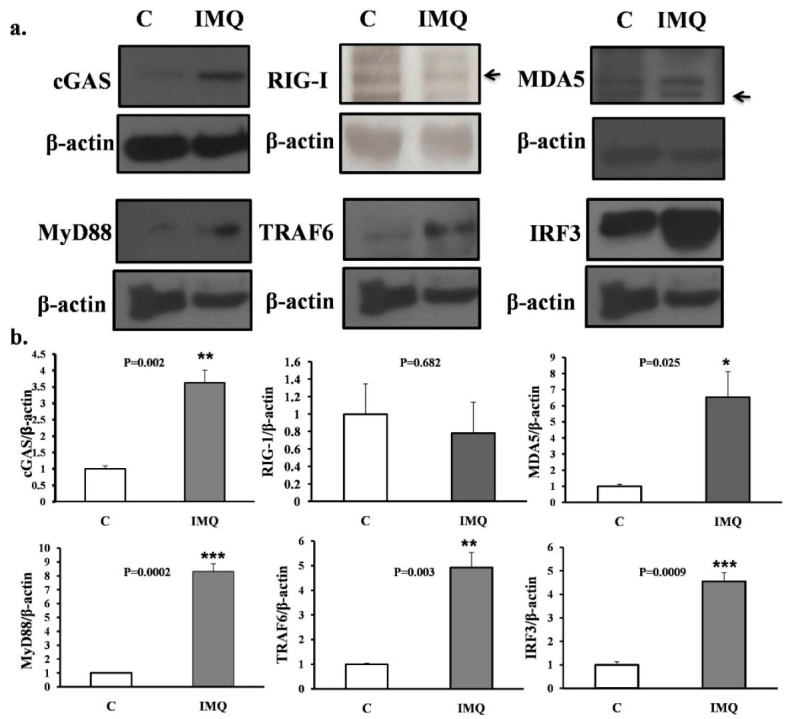
(**a**) Western blot results. (**b**) Histograms of intracellular proteins inside B cells. Western blot results of intracellular proteins inside B cells. Those B cells either from IMQ-stimulated or non-stimulated mice were co-cultured with CD40L feeder cells for 7 days and were then retrieved for analysis. Cytosolic protein was separated with the centrifuge method and blotted with antibodies against mouse cGAS, RIG-1, MDA5, MyD88, TRAF6, and IRF3, as shown in the upper part of the figure. The histograms are shown in the lower part of the figure. All the experiments were repeatedly performed at least three times and the comparison between groups, control (C), and IMQ-treated (IMQ), were calculated with the Mann–Whitney test. The mark * indicates a *p*-value less than 0.05. ** indicates a *p*-value less than 0.01. *** indicates a *p*-value less than 0.001.

**Figure 4 cells-11-03350-f004:**
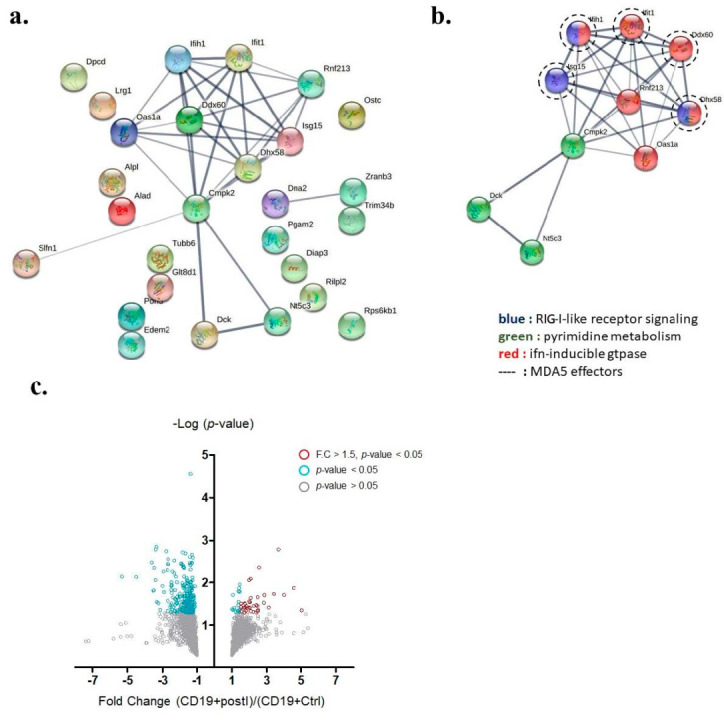
Key interactome of differentially up-regulated genes in auto-reactive B cells. (**a**) The protein −protein interactome of 27 up-regulated annotated genes was analyzed by the STRING database. (**b**) A key interactome containing 10 genes was discovered (Ifih1 gene encodes MDA5). (**c**) Volcano plot demonstrating proteins differentially expressed in B cells from either IMQ-stimulated or non-stimulated mice. Grey circles denote *p*-value > 0.05, blue circles denote *p*-value < 0.05, and red circles denote *p*-value < 0.05 and fold change greater than 1.5.

**Figure 5 cells-11-03350-f005:**
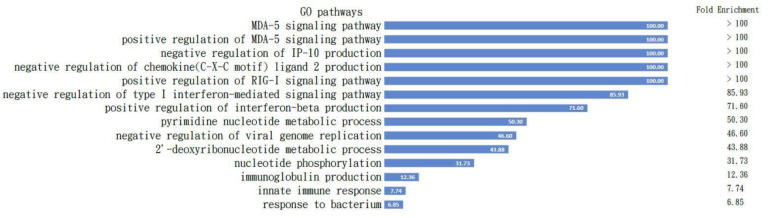
Pathways activated in B cells. All the proteins listed in Table 1 are comparable to the Mus musculus in the Panther classification pathway analysis and the results demonstrated positive regulation of the MDA5 signaling pathway with highest fold enrichment over 100; the pathways activated with these proteins are demonstrated. The fold changes are listed in the column on the right.

**Table 1 cells-11-03350-t001:** Proteins from B cells differentially expressed in between mice stimulated with and without imiquimod and positively selected with CD19+ surface markers and then with 1.5 times of change and with *p*-value < 0.05.

Accession	Gene	Description	Fold Change	*p*-Value	Compartment
Q64282	IFIT1	Interferon-induced protein with tetratricopeptide repeats 1	5.02	0.04	cytosol
A0A1Y7VJN6	IGHG3	Immunoglobulin heavy constant gamma 3	4.54	0.01	secreted
Q64339	ISG15	Ubiquitin-like protein ISG15	3.99	0.02	cytosol
Q8VI93	ALAD	2′-5′-oligoadenylate synthase 3	3.66	0.00	cytosol
P11928	OAS1A	2′-5′-oligoadenylate synthase 1A	3.42	0.02	cytosol
O70250	PGAM2	Phosphoglycerate mutase 2	3.12	0.04	cytosol
Q6NSU3	GLT8D1	Glycosyltransferase 8 domain-containing protein 1	2.86	0.02	cytosol
F8WIG5	DIAPH3	Protein diaphanous homolog 3	2.83	0.03	cytosol
A0A075B5R7	IGHV14-2	Immunoglobulin heavy variable 14-2	2.56	0.00	secreted
Q9Z0I7	SLFN1	Schlafen 1	2.51	0.04	cytosol
A0A140T8P2	IGKV6-20	Immunoglobulin kappa variable 6-20	2.48	0.05	secreted
A0A075B5M7	IGKV5-39	Immunoglobulin kappa variable 5-39	2.47	0.02	secreted
P01635	IGKV12-41	Ig kappa chain V-V region K2	2.45	0.03	secreted
Q8BPA8	DPCD	Protein DPCD	2.39	0.03	cytosol
Q91XL1	LRG1	Leucine-rich HEV glycoprotein	2.39	0.05	cytosol
P43346	DCK	Deoxycytidine kinase	2.24	0.04	cytosol
Q8BJT9	EDEM2	ER degradation-enhancing alpha-mannosidase-like protein 2	2.22	0.05	cytosol
Q8BSK8	RPS6KB1	Ribosomal protein S6 kinase beta-1	2.08	0.05	cytosol
Q3U5Q7	CMPK2	UMP-CMP kinase 2, mitochondrial	2.08	0.01	cytosol
Q99J87	DHX58	Probable ATP-dependent RNA helicase DHX58	2.08	0.02	cytosol
E9PZQ1	DDX60	DExD/H box helicase 60	2.02	0.03	cytosol
A0A171EBL2	RNF213	E3 ubiquitin-protein ligase RNF213	2.00	0.03	cytosol
Q62087	PON3	Serum paraoxonase/lactonase 3	1.95	0.04	cytosol
Q6NZP1	ZRANB3	DNA annealing helicase and endonuclease ZRANB3	1.85	0.04	cytosol
A0A140T8N3	IGKV13-84	Immunoglobulin kappa chain variable 13-84	1.85	0.04	secreted
Q9D020	NT5C3A	Cytosolic 5′-nucleotidase 3A	1.81	0.05	cytosol
Q922F4	TUBB6	Tubulin beta-6 chain	1.76	0.03	cytosol
Q8R5F7	IFIH1	Interferon-induced helicase C domain-containing protein 1	1.74	0.04	cytosol
P09242	ALPL	Alkaline phosphatase, tissue-nonspecific isozyme	1.74	0.03	cytosol
A0A075B5K6	IGKV2-109	Immunoglobulin kappa variable 2-109	1.71	0.05	secreted
Q99LE1	RILPL2	RILP-like protein 2	1.67	0.04	cytosol
Q78XF5	OSTC	Oligosaccharyltransferase complex subunit OSTC	1.53	0.03	cytosol
Q6ZQJ5	DNA2	DNA replication ATP-dependent helicase/nuclease DNA2	1.53	0.05	cytosol
J3QNR8	TRIM34B	Tripartite motif-containing 34B	1.52	0.04	cytosol

## Data Availability

All data are available upon reasonable request. Original proteomics data are available via ProteomeXchange with identifier PXD034855.

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
