# Peer review of "A Study on MDA5 Signaling in Splenic B Cells from an Imiquimod-Induced Lupus Mouse Model with Proteomics"

_cells, 2022, doi:10.3390/cells11213350_

Round 1
Reviewer 1 Report
This manuscript uses an established imiquimod-induced lupus mouse model and proteomics to reveal an MDA5 signaling pathway that is dysregulated in B cells from lupus mice, leading to increased anti-dsDNA. To this reviewer (who is not in the SLE field), what is novel about this work and what is learned is not stated.
For example, the introduction identified MDA5, RIG-1, and LGP as potential players in SLE. MDA is higher in SLE. MAVs is also presented as a regulated protein. However, how MAVs, MDA5, RIG-1 and LGP2 are linked to TLR7, which is the target of Imiquimod, is not obvious. In this reviewer’s opinion, the introduction needs to be revised to link TLR7 to the intracellular signaling of MDA5, Rig-1, and LGPs and maybe MAVs.
Methods could be improved.
One detail that is not obvious to a non-B cell scientist is why the B cells, isolated from animals with an SLE phenotype have to be cultured. Why is the experiment not done on freshly isolated B cells that are presumably contributing to disease?
How many times was the proteomics done? Was it done on samples pooled from multiple animals or from single animals?
Statistical analyses section should be extensively revised. How many animals were used for the histology? All the figure legends to include the number of experiments run, the statistical analysis that was done, and what the ** mean.
Figure legends need extensive revision to include the details needed to understand the figure without reading the results. A recommendation is that the title of the figure state the conclusions from that figure. At the least, the number of n’s and statistical analysis needs to be included for each figure.
For example (and this applies to most of the figures)
Figure 1: Figure 1a,c,and d are representative of how many animals? How many data points are contained within the bars in Figure 1b? Are the differences in Fig 1b significant? What am I looking at in Figure 1d? Indicate lymphocyte infiltration with arrows or by showing regions of inflammation in an enlarged view and indicate the inflammatory cells. Also, Figure 1c needs better labels so it is clear what the different blocks mean (presumably the top row is protein and bottom glucose but that should be clearly indicated on the image. (Took this reader some time to realize that the bottom panel is the standards for the top images).
Figure 2: B cells are isolated from the spleen. Is there is splenomegaly in the SLE animals? What is Figure 2a. Both treated animals? The same treated animal at different magnifications? Figure 1d, what is this experiment? Cultured cells from ± IMQ animals treated with feeder cells then the dsDNA analyzed (how?) from the media? Critical details are missing.
There seems to be something odd with Figures 4 and 5. Figure 5 seems to be the ranking of proteins from the Westerns that appear to be in Figure 4? Again, with no details in the figure legends, it’s difficult to follow the data.
The Discussion doesn’t really tie all the data together. It talks extensively about dsRNa, dsDNA but those aren’t the focus of the figures with only one figure quantifying secretion of dsDNA. Same with TLR9, not really dealt with in the figures. It also brings in the mitochondria which are not mentioned at all in the results. This reviewer felt that the discussion decreased the significance of the data by pointing out that others had seen MDA5, TLR7. This should be rewritten to highlight the new contributions of this work to our understanding of B cell dysfunction in SLE.
Author Response
Thank you for giving us an opportunity to revise our work. Please see the attachment.

Reviewer 2 Report
In this study, Yu-Jih Su et.al, utilized proteomics analysis to provide insights into intracellular signaling pathways that could propagate B cell activation and increase anti-dsDNA in an imiquimod-induced lupus mouse model, especially in response to viral insults.
Here are some of my concerns :
1- The methodology for disease induction, and experimental time points are unclear, please provide details besides of referring to previous work in reference # 21.
2- Immunohistopathological analysis of mouse kidney, liver, and spleen and Histology and immunohistochemistry are contradictory in regard to tissue thickness. Similarly, the author represented the histopathology approach as immunohistopathology and failed to provide details about what proteins they are staining for in the immunohistopathology section in line 133. Please provide detailed information about the antibodies that were used. Importantly, Both the methodology section and results are incompatible and I don’t see in the result section any immunohistochemistry finding..
3- Under section 2.4. Splenocytes cultured with feeder cells and detecting B cell profiles: authors referred that “Splenocytes were isolated and co-cultured with feeder cells” however they clearly indicated in the results that the cocultures were for purified B cells and feeder cells. So please correct this inconsistency. Also, what are the CD40L + feeder cells? were these from a cell line? if so please provide details. or were they isolated and characterized in your lab? if so please detail the method and validation strategy.
4- The result section is in need of reformatting and accurate representation. Here are some points:
- All the results are poorly written with the text missing figure panels or providing incorrect orders. Figure legends are not clear and lack important information such as the number of animals used, time points for each experiment, number of experimental replicates, etc.
- All results are missing quantitative data representation and just providing pictures. For example:
- Figure 1a: missing a scale.
- Figure 1C: should be ordered B: quantitative group analysis is needed beside this picture?
- Figure 1b the slight increase in cre, BUN does not indicate the establishment of the disease. what is the time point for these measures and how it looks like over time? also, figure B is not fully described in the text. Similarly, quantitative data analysis for the picture is needed
- Figure 2 a :c pictures are not satisfactory and need quantitative stats for description.
- What is the ELISA kit used please describe the material and methods
- Also I am assuming these are in the culture supernatant, how many experimental replicates are represented?
- Figure 5 is not correctly inserted in the text. Also, How many experimental replicates are presented in WB? To me, the B actin seemed to be all to one sample.
Author Response

(The authors gave the same response as above.)

Reviewer 3 Report
The authors present a study on MDA5 signaling in splenic B cells of mice with imiquimod induced lupus. Overall the subject is important. However, there are some main issues with the manuscript that should be addressed.
1. The Introduction should be more informative. The arguments for the research are unclear.
2. There is no description of control and untreated groups of mice. Animal breeding conditions were not indicated.
3. Splenocytes isolation method and flow cytometry gating strategy needs to be presented.
I have made specific comments below:
Introduction
Line 42-43: The sentence: Antigen-presenting human leukocyte antigen molecules are identified in various ethnic groups is not clear to me. I undersand the autors wish to say: HLA molecules play a crucial role in antigen presentation and genetic diversity is observed.
Line 61-65: The sentences should be more informative.
Line 89-93: I am not sure why authors mention about COVID-19. There is no more information about COVID-19 in the manuscript. The argument for this is convoluted.
M&M
Line 95-102: The lupus mouse model induction should be clearly described.
Line 124: Method of splenocytes isolation should be presented.
Line 131-134: What kind of tissue does the description apply to?
Results
Pointing arrows should be added to Figure 1D.
The name of section 3.1. and 3.2 shoud be more informative. E.g. B cells preparation from fresh splenocytes via negative selection method is not a description of the results.
Author Response

(The authors gave the same response as above.)

Round 2
Reviewer 1 Report
Results are disorganized.
Present the figures in the same order they are described in the results.
Methods: Were male and female mice used?
Line 258: Define GPT
Lines 289-290. Fig 2b needs to be explained.
Figure 2C is not a photo, result needs to be given in text.
The text does not follow Figure 2. Start with Fig 2a, describe that, then move to fig 2b, 2c, etc.
LIines 293-294 How do the results in demonstrate functional differences?
Figure 3 needs to be described before Figure 4. Or switch Figures 3 and 4.
Author Response
Dear Reviewer,
Enclosed you will find our replies to your comments.
